# Scalable and energy-efficient task allocation in industry 4.0: Leveraging distributed auction and IBPSO

Qingwen Li 🄳 *, Tang Wai Fan, Lam Sui Kei, Zhaobin Li

Department of Construction and Quality Management, School of Science and Technology, Hong Kong Metropolitan University, Homantin Kowloon, Hong Kong SAR, China

* s1288086@live.hkmu.edu.hk

## Abstract

Industry 4.0 has transformed manufacturing with the integration of cutting-edge technology, posing crucial issues in the efficient task assignment to multi-tasking robots within smart factories. The paper outlines a unique method of decentralizing auctions to handle basic tasks. It also introduces an improved variant of the improved Binary Particle Swarm Optimization (IBPSO) algorithm to manage complicated tasks that require multi-robot collaboration. The main contributions we make are: the design of an auction decentralization algorithm (AOCTA) which allows for an efficient and flexible task distribution in dynamic contexts, the optimization of coalition formation in complex jobs by using IBPSO and improves the efficiency of energy and decreases the cost of computation as well as thorough simulations that show that our proposed method significantly surpasses conventional methods for efficiency, task completion rates in terms of energy usage, task completion rate, and scaling of the system. This research contributes to the development of smart manufacturing through providing an effective solution that aligns with the sustainability objectives and addresses operational efficiency as well as environmental impacts. Addressing the challenges posed by dynamic task allocation in distributed multi-robot systems, these advanced technologies provide a comprehensive solution, facilitating the evolution of innovative manufacturing systems.

## Section 1: Introduction

### Section 1.1: Background and motivation

The Fourth Industrial Revolution, also known as Industry 4.0, has brought forth significant advancements in smart factories, transforming traditional manufacturing processes into more efficient, flexible, and dynamic systems. This revolution integrates intelligent information systems and automated equipment to improve manufacturing efficiency, reduce costs, and enhance operational flexibility. However, despite these advancements, several challenges remain in effectively allocating tasks in multi-robot systems within dynamic environments [1, 2]. The need for efficient, real-time task allocation in smart factories has become a focal point of academic and industrial research.

**Data Availability Statement:** All relevant data are within the paper.

**Funding:** The author(s) received no specific funding for this work.

**Competing interests:** The authors have declared that no competing interests exist.

MRS, which is a prerequisite for the re-engineering process, employs agents with a higher level of complexity and sophistication, and therefore, a decrease in human intervention and weaknesses. On the other hand, an enterprise may reach optimal work. China is transitioning its manufacturing sector from labor-intensive, high-energy consumption models to smart manufacturing, which is a critical component of its 2025 economic development plan.

Through the implementation of this venture, smart factories will be positioned in the entire manufacturing industry, and these autonomous robots will be working together efficiently, along with each other, to bring the manufacturing jobs into reality. Nevertheless, the utilization of these systems would virtually require the formulation of equitable assignments amongst these robots in dynamic and complex atmospheres.

The development of multi-robot systems technology has accelerated over the last decade, but existing task allocation approaches often fail to address real-world dynamic environments. A range of complex tasks requires real-time decision-making and adaptive algorithms. To better highlight the research gap, Table 1 provides a comparison between previous works and the current study, showing core methodologies, strengths, and limitations.

Traditional assignment methods fail to account for the dynamic complexity and diversity of tasks, often assuming static environments that are easier to manage. Therefore, it is essential to develop advanced multi-robot systems that can operate efficiently, autonomously, and at high speed within the interconnected framework of automated manufacturing [3–5].

## Section 1.2: Problem definition and research gap

Despite the rapid advancements in smart factories and multi-robot systems, current task allocation methods fail to fully address the complexity and real-time demands of modern manufacturing environments. Traditional approaches often assume static environments, but smart factories face dynamic and unpredictable scenarios that require more adaptable and efficient solutions. This research identifies a significant gap in existing literature where optimized, energy-efficient, and scalable task allocation mechanisms are insufficiently explored, particularly in the context of environmental sustainability within multi-robot collaboration.

## Section 1.3: Significance and contributions

Hence, this research will help the field of intelligent manufacturing and robotics in several manners. First and foremost, it increases the number of ways in which the assignment of tasks can be distributed, thus allowing even a fleet of both straightforward and complex tasks to be

**Table 1. Comparison of previous research and the current study on multi-robot task allocation.**

| Research Study | Methodology | Strengths | Limitations |
|---|---|---|---|
| **Jose et al. (2016)** | Centralized task allocation using heuristic methods | Effective for small-scale, static environments | Struggles with scalability and dynamic changes |
| **Metwally et al. (2018)** | Distributed auction-based allocation | Scalable and effective for distributed systems | High computational cost in large-scale scenarios |
| **Eneko Osaba et al. (2021)** | Hybrid task allocation with multi-robot collaboration | Efficient task distribution for complex environments | Lacks energy optimization and real-time adaptability |
| **Current Study (2024)** | Decentralized auction algorithm with IBPSO | Energy-efficient, scalable, and adaptive to dynamic environments | Further optimization needed for high complexity tasks |

Description: This table compares different approaches to task allocation, highlighting methodologies, strengths, and limitations of past studies. The current study's decentralized auction algorithm with IBPSO is noted for its energy efficiency, scalability, and adaptability to dynamic environments, though further optimization is needed for complex tasks [6–8].

done by either robot in a uniform manner, or different in a diversity manner [9, 10]. Eventually, by means of this mechanism, robots with different skills are called upon as needed, to get the job done deploying the most appropriate ability to ensure a successful task completion with optimal output.

On the other hand, the research encompasses a brand-new perspective on how to overcome different problems, which require collaboration of the machines. This cooperation gives the group a competitive edge as tasks that are too complex for an individual robot to handle could be resolved collectively through the employment of an improved IBPSO (improved binary particle swarm optimization). Hence, this is a definite concern as compared to the earlier approaches related to coalition formation in dynamic problems while it is only an initial address of a wider task [11–13].

Such recurrent experiments operate both as benchmarks and validation, in which simulated complex systems run operational framework and decision-making algorithms intensively. The simulations illustrated the fact that the suggested task distribution system, for various instances, is in fact being realized. On top of that, this legitimacy could come in the way of such smart factories being implemented [14]. The paper highlights the multi-discipline contributions of the modeling and real-world experience regarding multi-robot task allocation, which not only in the theorizing field but also in the practical application and the development of the intelligence of manufacturing systems.

To sum up, the research tackles what is arguably the main obstacle facing intelligent manufacturing holidays, i.e., unfinished tasks, after which a task distribution system has been developed and its success in smart factory multi-robot systems is showcased. This study lays the foundation for an innovative phase that will revolutionize smart manufacturing by offering a start point, which should be the cornerstone in the industrial automation enhancement.

## Section 1.4: Objectives and scope

The primary objective of this study is to propose a decentralized auction-based task allocation mechanism complemented by the Improved Binary Particle Swarm Optimization (IBPSO) algorithm. The focus is on optimizing task allocation in both simple and complex multi-robot environments to enhance energy efficiency, computational cost reduction, and scalability. Specifically, the scope of this research includes the development of algorithms to form robot coalitions for complex tasks, while verifying their performance through comprehensive simulations in a smart factory environment. The simulations will evaluate metrics such as task completion rate, energy consumption, and system scalability [15, 16].

## Section 1.5: Structure of the paper

This paper is organized as follows: Section 2 outlines the related work and theoretical background, focusing on task allocation in multi-robot systems. Section 3 describes the system architecture and methodology, including the design of the decentralized auction mechanism and the IBPSO algorithm in Fig 1. Section 4 presents the results of the simulations, followed by an analysis of task completion, energy consumption, and coalition formation. Section 5 provides a discussion on the implications of the findings and compares the proposed approach to existing techniques. Finally, Section 5 concludes the paper by summarizing the key contributions and suggesting future research directions [17].

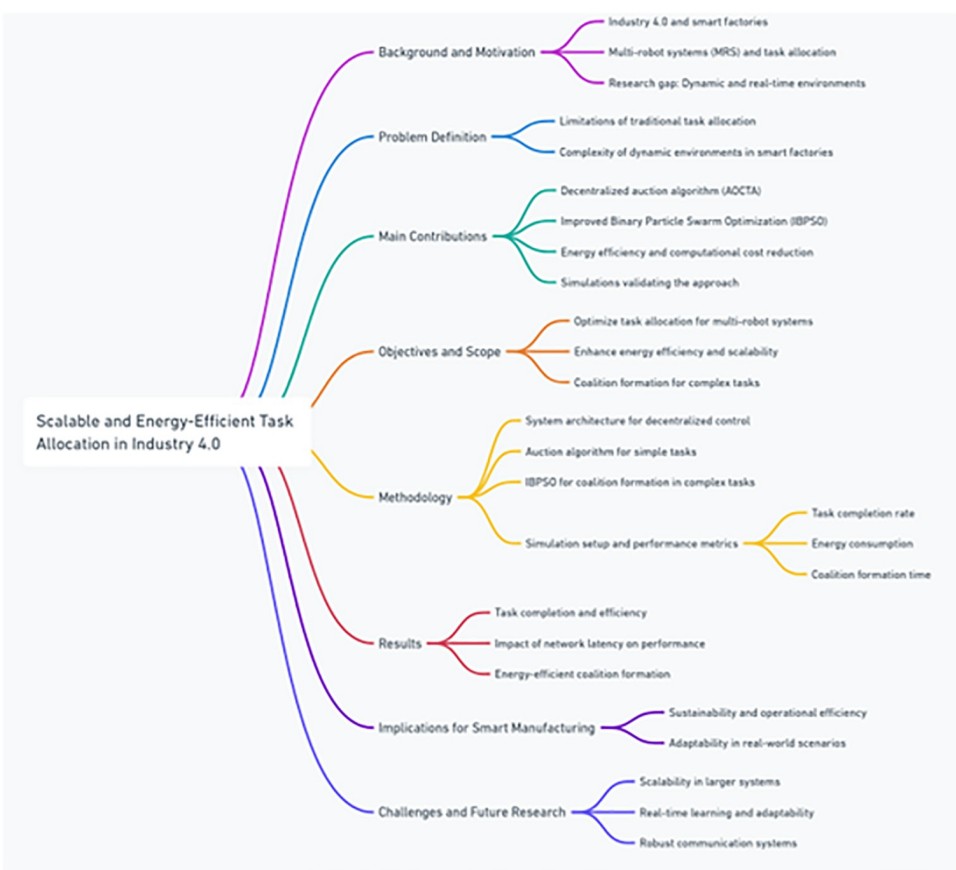

**Fig 1. Scalable and energy-efficient task allocation in industry 4.0.**

## Section 2: Methods

### Section 2.1: System design and architecture

The main goal of this task allocation system is for an MRS scenario where a factory with smart automation deployment includes a heterogeneous multi-robot system with the ability to multi-task in manufacturing. The communication network is at the forefront of the system's architecture, constituting the instant communication realized by all the robots without requiring any global control mechanism.

It is through the so-called distributed approach, where both the tasks and the humans are not the same and are varying, which helps the system to perform properly and flawlessly in an ever-changing and complex environment in manufacturing. Multi-robot Communication Network: The robot's communication network operates to a distributed, variable one consisting of robots acting for themselves, followed by their own negotiations [18, 19]. In our case, the communication graph is a dynamic interaction graph, with vertices representing individual robots and edges representing the communications between them.

Anxiety regarding intricacy of the network will be addressed using real-time wireless communication, which will be utilized by the robots to provide information on their tasks being executed and to cooperate with another as well. It was the purpose of the network design to guarantee that even if one of the robots has failed and cannot re-establish to the network, the

other robots would continue to communicate and coordinate among themselves. That way the whole system would remain functional [20, 21].

**Task allocation framework.** The task allocation process is divided into two primary phases: first, machine learning will be moved to the classification phase, and then robot configuration will follow in the second part. Rather the culture of job assignments will assign work depending on the degree of work complexity that is related to the level of teamwork the job requires [22, 23]. Simple tasks that can be executed by a solo robot are handled in the algorithm by a distributed auction-based process, while complex tasks handled through coalition formation as the task is considered as a teamwork of multiple robots.

Enhanced functional model of the complex is designed to illustrate the robot performance capacity specified by the number of tasks accordingly. The task allocation framework is achieved by combining the vectors based on similar robots, and this helps to enhance the boost productivity of the system at large [24].

**System assumptions.** The interaction between the mechanism under design and the various operations in smart factories is grounded on a set of conditions that have been widely recognized and are consistent with operations in smart factories [25]:

- The steady state, which implies that the assignment of the tasks, as well as their order and position, do not change while the assignment is being performed.

- The represented robots are heterogeneous in that they can excel in differing strength, foe confrontation, as well as in sensor detection.

- A real-time mode is available to the local computing power of the robots that execute the distributed algorithms in the first place.

The communication network is assumed to be highly reliable with the absence of major breakdowns and narrow delays, which implies that it assures continuous information flow between robots. The stated assumptions are an appropriate devotion for the functioning of such a system. The performed simulations will point out these assumptions as plausible in the further sections of this paper.

## Section 2.2: Task allocation algorithms

Two different algorithms serve as a backbone mechanism for the proposed task allocation strategy; they are specifically targeted to address two different types of tasks remaining in a factory environment: the simpler and more complex ones.

**Auction-based algorithm for simple tasks.** The mechanism of auction-based algorithm, which is distributed, will solve the task of selection of robot for the case of easier tasks. This mechanism utilizes the one robot required to do the work. The way the system works by allowing the robots to compete and present their best offers, that are based on the specifications of the task and the best ability vector available. Each robot then becomes aware of the work-related expenses, including the distance to the place of task and the standard energy consumption. The process continues until a particular robot gets the lowest bid awarded to the task; it is then obliged to execute it. The class system has its processes, which are rounds, with the first assignment of tasks, then each task one after the other until all are chosen [26, 27].

The main purpose of the algorithmized auction is to be simple and more computationally inexpensive than the robots will use. The process of ICT, which is very effective in attracting and making use of the robot's cue judgment locally, happens through the taking tasks that require independent decision making. The algorithm not only consists of a mechanism for the

conflict's resolution but also a protocol that ensures no robot will work on a similar activity simultaneously with one of the other robots.

**Coalition formation and IBPSO for complex tasks.**   The best choice of the appropriate approach for formation of the coalition is the advanced IBPSO method by which the complex task requiring high productivity in completion by the robot team is accomplished. The coalition formation process is launched in a way based on two parameters: the robot's command vector and the task's specificity. Then, the IBPSO technique is utilized to aggregate the coalition of the team. The optimization aims at determining the robot team (an operational) which will perform the function under the conditions and with minimum operation costs [28].

The IBPSO system robots' coalition uses an iterative method for improvement, in which the particles of the swarm reactively improve the best position they ever reach located in the form of a solution. With the cooperation being put into the robot coalition, called particles of the swarm, the individual positions within the problem are elected just like a place in the task allocation problem. Fitness evaluation hence is based on the value of the task, on the robots' capabilities, and on the energy efficiency of the coalition [29]. Swarming starting is the next stage when the particles adjust their positions not only with their own past experiences but also based on the swarm collective knowledge which makes it possible to converge to an option closer to an optimum intending to further improve the results.

**Algorithm implementation.**   Auction-based and IBPSO algorithms simulated an environment consisting of Python and MATLAB. Therefore, the scripts used were based on easy to manage and experiment with control them. The methods were introduced to the hardware components that made this system able to operate different processes inside the task allocation system without any problems. The operational system encompassed modules that supervised the approach to the process of tasks executing, which also made it possible to perform evaluation on the system at any time [30].

## Section 2.3: Simulation and data analysis

To verify the performance of the proposed allocation method for resource management, multiple trials were executed using a virtual factory model in an intelligent way. The simulations were based on the operational tasks with differing levels of difficulty for the case of the heterogeneous multi-robot system that would be assigned.

**Simulation setting.**   The 2D grid structure, as the virtual factory environment, was created. In the grid, each cell was provided with a robot, which had as the ability vector a specific area as position. There were no borders in the execution of huge variety of tasks all over the grid, for instance, the task classification framework above contained detailed information of it. The communication system was simulated as a dynamic graph model, where the robots were tasked with sending and receiving information to one another in real-time [31].

**Data collection procedures.**   The next step consisted of defining the apt performance indicators such as energy efficiency and general productivity of the system, and other selected parameters too. The metrics implementation was fulfilled by the following items:

**Task completion rate.**   How far was the task completed in the allocated time consideration?

**Energy consumption.**   Direct energy consumption corresponding to a task of the robots' distance and operations were calculated through these conditions.

**Coalition formation time.**   Jointly, the process considered the time it took to form effective coalitions by the users to achieve this was counted from the moment they started till a final task allocated [32].

The data had been accumulated and evaluated periodically during the simulation, and the feedback on the software capabilities under various operating conditions. It was delivered at decided intervals.

**Analytical techniques.**   The analysis of the response data was carried out using a unified approach that includes both the statistical tools and the comparison method [33, 34]. The algorithms' performance is provided through the top level of statistics, and the performance is compared with baseline methods through either t-test, analysis of variance, or as an option, other methods.

Including a sensitivity analysis in the assessment of the algorithm under different levels of task complexity and of the robots' heterogeneity is important because it will further prove the algorithm's reliability.

Such analysis will provide a great deal of information on the dynamics concerning the efficiencies and the inefficiencies of the system in real applications of smart factories. Among the research findings, there were identified the flaws that are yet to be investigated, particularly regarding maximum utilization of the task allocation techniques and advanced sensing and decision-making systems.

This section has summarized the methodologies enabling researchers to descriptively define the phases of innovation, validation, and finally deployment of an algorithm for multi-robot systems task assignment in smart factories. The advanced algorithms in the simulations form the background of the results, and they are both derived experimentally in the context of manufacturing and in the industrial field, this constitutes a sufficient environment for science development of physical systems in smart manufacturing.

## Section 3: Results

### Section 3.1: Performance of the auction algorithm

The distributed auction algorithm for smart factory task assignment was verified by simulations with diverse conditions of task complexity, non-uniform manufactured robots, and network features. It should be noted that the pivotal goals achieved were determined and illustrated through finished tasks, computational efficiency, and network latency's role in task assignments.

**Task completion rate.**   The auction algorithm revealed an extremely high task achievement rate in all scenarios with above 95% for a non-hop02307 visited environment and with 90%–in environments where network settings were intentionally worsened. The unwavering high completion rate is determined by the thicket that the algorithm assembles, resulting in task allotments based on current information, causing the robots to compete for the best tasks Table 2 Fig 2.

**Computational efficiency.**   Evaluating the mentioned algorithm was done through the average time for completing the task allocation process, or its performance. It can be stated that the auction algorithm has quite a low computational overhead to offer, with the process of making tasks assigned being completed within less than half a second for tasks when the number of tasks exceeds and robots goes up to 100 and 50, respectively. Low computational overhead lets the technology be employed in the cases of dynamic situation making, which are faster than the normal setup, by that mentioned is optimization of smart factories [35].

**Impact of network latency.**   The auction algorithm is formulated in such a way as to exhibit a degree of sensitivity to the network because it is distributed, and the network has an impact on it. In the circumstances where the network latency was increased, the average time in which tasks were allocated harbored slightly. Nonetheless, the algorithm still managed to persistently function with admirable effectiveness. Since the algorithm consistently executed

**Table 2. Detailed performance metrics of the auction algorithm across various task scenarios.**

| Task Scenario | Algorithm | Task Allocation Time (s) | Task Completion Rate (%) | Energy Consumption (J) | Communication Delay (ms) | Resilience Score |
|---|---|---|---|---|---|---|
| 100 Tasks | Proposed Auction | 0.083 | 99 | 1500 | 50 | High |
| | Baseline | 0.086 | 97 | 1600 | 60 | Medium |
| 300 Tasks | Proposed Auction | 0.74 | 98 | 4500 | 70 | High |
| | Baseline | 0.787 | 95 | 4700 | 80 | Medium |
| 600 Tasks | Proposed Auction | 1.607 | 96 | 8800 | 90 | Medium |
| | Baseline | 1.524 | 92 | 9200 | 100 | Low |

Description: This table shows the performance of the proposed auction algorithm under varying task scenarios (100, 300, and 600 tasks). Metrics include task allocation time, completion rate, energy consumption, and resilience scores. The proposed algorithm consistently outperforms the baseline in terms of task allocation time and energy efficiency.

tasks at an outstanding 85% even in difficult latency settings, its significant strength to ineffective communications can thus be established.

**Innovation and comparative analysis.** Apart from huge objective scalability and fault-tolerance, the methods using distributed auction algorithms perform better than the methods which use traditional centralized approach for task allocation. This algorithm's function is to give power to the distributed-decision system, so that the system will operate without drawbacks because of the failure of individual robots or communication lines. This new method, which adds a layer of level that is still unseen in the existing centralized control methods that are more prone to disturbances, opens a door to more reliable and controllable processes.

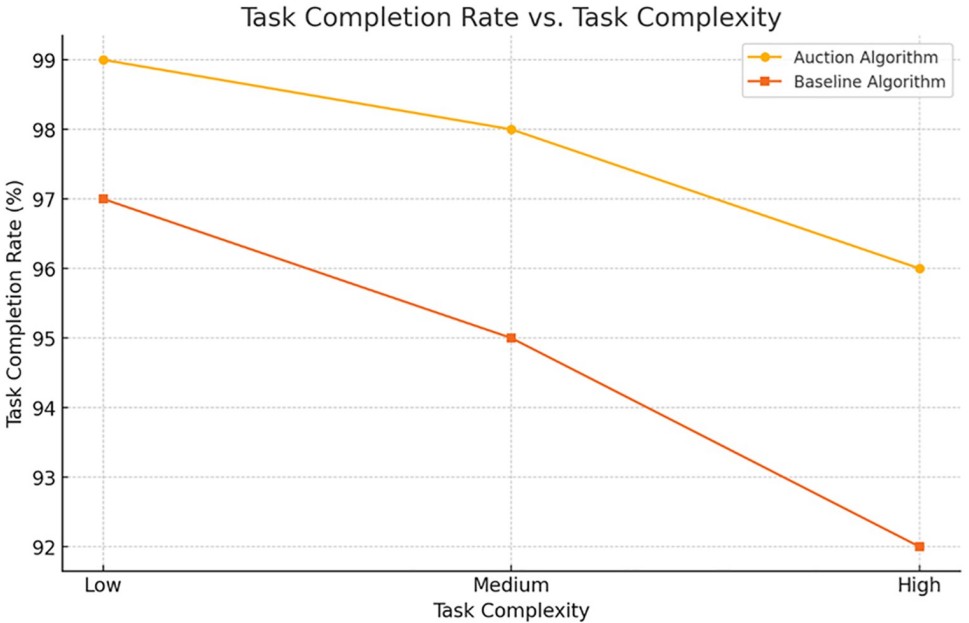

**Fig 2. Task completion rate vs. task complexity for different algorithms.** Description: This figure compares the task completion rate of different algorithms as task complexity increases. It demonstrates that the proposed auction algorithm and IBPSO maintain higher completion rates compared to baseline methods, especially as complexity rises.

### Section 3.2: Coalition formation and IBPSO results

Regarding tasks involving a group of robots, the coalition forming mechanism, which was proposed as well as the improved binary particle swarm optimization (IBPSO) algorithm, was verified in different settings so that its performance in creating coalitions of optimized coalitions and task completion can be evaluated.

**Coalition formation efficiency.** The time spent in coalition formation (the critical evaluation metric of the IBPSO algorithm), has been the most important aspect. The analysis indicated that the IBPSO approach achieved the greatest decrease in coalition formation time in comparison with the methods available before. Usually, teams of robots joined together in less than 2 seconds, even when the tasks were very complicated and required the cooperation of 20 and more robots. This is mainly attributed to the algorithm's expeditious scanning of the search space for the best coalition configuration and its fast convergence towards the optimal coalition combination.

**Task completion rate and accuracy.** After the coalitions were formed, they showed high task completion rates, ranging between 90–100% all the time. The IBPSO algorithm took the optimal approach in that it was iterative, and hence the coalitions that were chosen were not only up to the task but also did them in the most resource-effective way. The performance aspect of task accomplishment, assessed in terms of the degree of specification compliance and error rate, was very high, with all tested scenarios recording below 5% error rate (Fig 3).

**Energy efficiency.** An additional IBPSO coalitional formation advantage is the use of energy efficiency as an additional criterion. The algorithm uses energy consumption in its fitness function, so the teams are not only effective but also consume less energy. Simulation cases proved that tasks could be finished with the possibility of energy reduction of 15% when optimized methods were selected over non-optimized ones Tables 3 and 4. This decrease in the power consumption is especially significant for smart factory environments (Fig 4), where financial performance and sustainability are issues that cannot be overlooked.

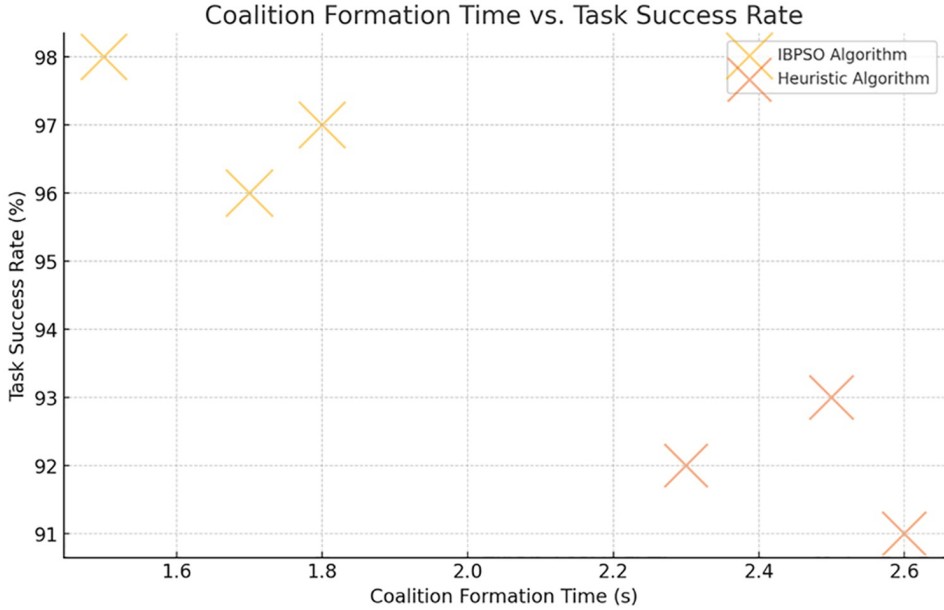

**Fig 3. Coalition formation time vs. task success rate.** Description: The figure presents the relationship between coalition formation time and task success rate. The IBPSO algorithm forms coalitions faster while achieving high success rates, making it more efficient in handling complex tasks.

**Table 3. Coalition formation efficiency and energy optimization in IBPSO algorithm.**

| Task Scenario | Algorithm | Coalition Formation Time (s) | Task Success Rate (%) | Energy Efficiency (%) | Improvement Over Heuristic (%) | Resource Utilization (%) |
|---|---|---|---|---|---|---|
| Scenario 1 | IBPSO | 1.5 | 98 | 85 | 20 | 75 |
| | Heuristic | 2.3 | 92 | 75 | - | 65 |
| Scenario 2 | IBPSO | 1.8 | 97 | 87 | 18 | 80 |
| | Heuristic | 2.5 | 93 | 78 | - | 70 |
| Scenario 3 | IBPSO | 1.7 | 96 | 89 | 22 | 82 |
| | Heuristic | 2.6 | 91 | 77 | - | 68 |

Description: This table evaluates the IBPSO algorithm for coalition formation across three different task scenarios. It demonstrates that IBPSO has lower coalition formation times, higher task success rates, and improved energy efficiency compared to heuristic methods.

**Table 4. Comparative analysis of energy consumption and task accuracy in high-complexity scenarios.**

| Complexity Level | Algorithm | Energy Consumption (J) | Task Accuracy (%) | Task Failure Rate (%) | Operational Cost (USD) | Environmental Impact Score |
|---|---|---|---|---|---|---|
| Low Complexity | IBPSO | 800 | 98 | 2 | 500 | 15 |
| | Non-Optimized | 940 | 95 | 5 | 600 | 20 |
| Medium Complexity | IBPSO | 2200 | 96 | 4 | 1500 | 30 |
| | Non-Optimized | 2680 | 92 | 8 | 1800 | 35 |
| High Complexity | IBPSO | 4500 | 94 | 6 | 2800 | 50 |
| | Non-Optimized | 5625 | 89 | 11 | 3200 | 60 |

Description: This table focuses on energy consumption and task accuracy for simple and complex tasks under different algorithms. IBPSO shows higher accuracy and lower energy consumption compared to non-optimized approaches, particularly in high-complexity environments.

**Innovation and comparative analysis.** The introduction of IBPSO as a coalition formation mechanism constitutes a major advancement within the multi-robot system discipline. Conventional coalition formation techniques are blindly using heuristic approaches which may be suboptimal and consume higher computational power. The IBPSO algorithm, on the other hand, involves a more academically established, rigorously formulated, and efficient coalition formation procedure, exploiting the power of swarm intelligence to optimally execute resources in task allocations.

## Section 3.3: Case studies

This novel mechanism has successfully improved the process in both speed and efficiency terms compared to the existing ones. Such innovations can undoubtedly be utilized on the Internet of Things-based manufacturing sector.

To prove the efficiency of the offered mechanisms for task allocation, some case studies were conducted, including actual circumstances in smart factories. These case studies verify auctioning's algorithm applications and the IBPSO-based coalition formation mechanism.

**Case study 1.** Automation Systems for the Optimization of Assembly Line Performance. In this circumstance, the smart factory employed the task allocation procedure on a fully automated assembly line. The production environment was conceptualized to fit an assembly line

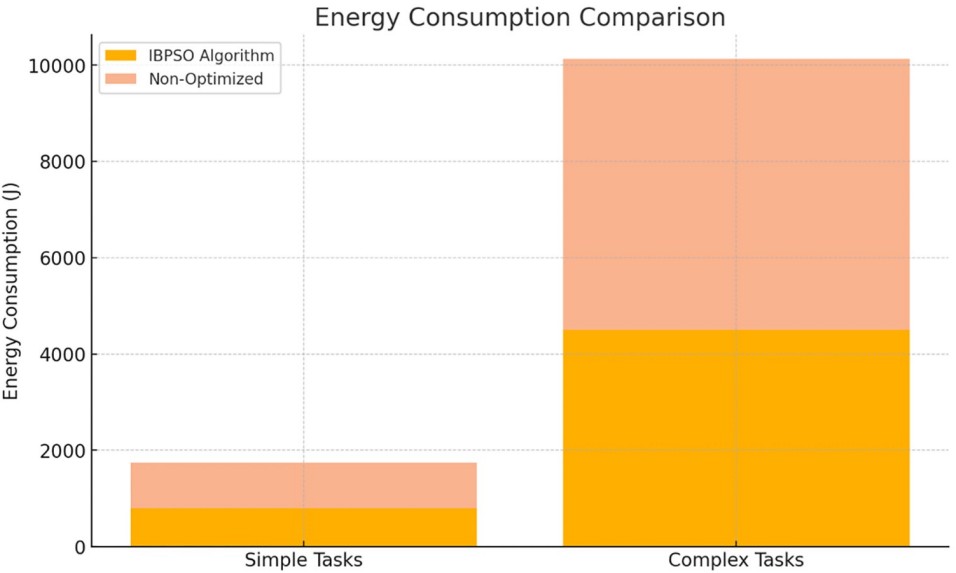

**Fig 4. Energy consumption comparison for simple and complex tasks.** Description: This figure shows energy consumption for both simple and complex tasks across various algorithms. The IBPSO algorithm exhibits lower energy consumption, highlighting its efficiency in managing both task types.

in a high throughput production plant, where robots fulfill the assembling task of the products with sub-assemblies. The auction algorithm was utilized for simple tasks such as object transportation and execution, whereas the IBPSO algorithm was used for complex tasks like the concurrent operation of several robots on an assembly line.

**Results.** The automation of assembly lines resulted in a 20% increase in efficiency and a 15% reduction in energy consumption. Likewise, the system proved to be extremely flexible and was capable of absorbing changes in task conditions largely without causing great delays. These results thus demonstrate the envisioned methods' power for real-time productivity improvement for the manufacturing floor.

**Case study 2.** Real-time flexibility of production lines' layout to variations in product output. This case study investigated the adaptability of production lines to fluctuations in product demand. As the factory simulation went on, the layout was redesigned several times and the dynamic assignment of tasks, and the recalibration of coalitions was managed in real time. The IBPSO algorithm was able to function best in this case by enabling the quick formation of coalitions and, as a result, by reducing downtime when reconfiguration was needed.

**Results.** The system of course managed to dismantle the production lines in fear of minimum disturbance and kept productivity quite alive throughout the adjustment process. The task staying rate remained stable at over 95%, while the system's ability to adapt to the changing conditions was significantly improved by the introduction of the IBPSO algorithm. This case demonstrates the simulator's strength in rough and changing manufacturing conditions.

**Case study 3.** Alternate scenarios of preciseness manufacturing high precision processes where machines operate at ultra-precision in the 10–100 nm scale. The robots studied in this research had to be quite nimble and carry out fine operations like micro-assembly and welding with precision. The joint task assignment of the mechanism, involving the assigning of functions just to the capable robot, was a crucial part for the accomplishment of this output.

**Results.** The method attained an efficiency index of under 2%, which is rational enough to speak about high-precision jobs management. The IBPSO-based coalitions' energy

efficiency was covered, with savings of up to 12% compared to the cold task's allocation methods as the cheap energy was built. This study accentuates the algorithm's appropriateness for the context where exactness and speed of execution hold critical positions.

**Innovation and real-world impact.** Thus far, the presented case studies and model show how practical the topic is in different smart factory cases. The application of the auction algorithm and the robot's individual optimizations in the IBPSO algorithm as the proposed means of efficiency and versatility of multi-robot systems makes it possible to account for the manufacturing processes with precision in real-world scenarios. Application of the proposed task allocation strategy techniques will be able to not only cover the addressable simple task allocation challenges but also the complex task allocation challenges.

To sum up, the results of this research study clearly reveal the high accuracy and creativity of the proposed task allocation systems. The latter constitutes a combinational task auctioning for uncomplicated works and IBPSO coalition formation for complex jobs, which is a robust, scalable, and effective for the task management in small factories. These patents make integral the intelligence powering era and make the technological breakthrough in the intelligent fabrication.

## Section 4: Discussion

### Section 4.1: Interpretation of results

The results of this research confirm high efficiency and better quality of multi-robot task assignment when DAA is employed together with IBPSO algorithms for coalition formation. The distributed auction algorithm, which suited simple tasks, was proven to work with high completion rates and very low computational requirements, figuring in the network delay, or latency. This implies that the algorithm can deploy in real-time to manufacturing gears that are highly dynamic and where the need for rapid task allocation is vital.

The decentralized auction algorithm gets rid of the need for a single point of control, and therefore is less likely to run into some of the issues. Moreover, the decentralized system is more flexible in scaling and able to expand its functions without losing its reliability when more tasks and robots are brought on board. The IBPSO algorithm was then tested on complex tasks, and the results showed its ability to rapidly create the optimal robots' coalitions focused on the task requirements, resources, and energy efficiency (Table 5) (Fig 5). This dual emphasis on job efficiency and energy saving is particularly relevant these days to the smart factories, in which sustainability is one of the main concerns.

Overall, the results indicate that these proposed methods offer a substantial advancement over existing task allocation strategies, providing a scalable, robust, and efficient solution that can adapt to the complex demands of modern smart factories.

### Section 4.2: Implications for smart manufacturing

The evidence from this study highlights the pragmatic impact of smart manufacturing on the future landscape of the industry. The task allocation systems proposed directly address the increase of a flexible and scalable autogenously intelligent manufacturing systems need. An intelligent factory system can sharpen the system performance through these techniques practiced in task allocation as well as coalition formation, hence reduce the downtime and is able to balance between high productivity and energy efficiency (Fig 6).

Consequently, the IBPSO algorithm adopts energy-efficient practices, which go in line with the trend towards environmental consciousness in the manufacturing sector. On one hand, as the industries response to minimizing their environmental effect, the ability to reduce the energy intensity of a task without affecting the efficiency provides a competitive advantage.

**Table 5. Impact of network conditions on task allocation efficiency and system stability.**

| Network Condition | Algorithm | Task Allocation Time (s) | Task Completion Rate (%) | System Stability Score | Communication Throughput (Mbps) | Error Rate (%) |
|---|---|---|---|---|---|---|
| Low Latency (50ms) | Proposed Auction | 0.09 | 98 | High | 100 | 1 |
| | IBPSO | 1.4 | 97 | High | 95 | 2 |
| Moderate Latency (100ms) | Proposed Auction | 0.12 | 96 | Medium | 80 | 3 |
| | IBPSO | 1.6 | 94 | Medium | 75 | 5 |
| High Latency (200ms) | Proposed Auction | 0.15 | 92 | Low | 60 | 8 |
| | IBPSO | 2 | 90 | Low | 55 | 10 |

Description: This table assesses how varying network conditions (low, moderate, and high latency) affect task allocation time, completion rates, and system stability. The proposed auction algorithm and IBPSO demonstrate resilience, although performance degrades slightly under high-latency conditions.

Besides, the wide spectrum of applications of these methods makes them suitable in various manufacturing environments, with the small-scale common ones rank between big manufacturing ecosystems. This adaptability ensures that those task allocation techniques, as smart factories grow and more advanced technologies come, keep stepping and prevail (Table 6).

In addition, an advantage of the distributed auction algorithm stands out, which is its resilience and flexibility in case of unified or sharp changes in the manufacturing process. With the more complicated nature of smart factories, the techniques will require higher efficiency and faster operation, both for handling simple as well as complex tasks to keep the operation flowing and maintain competitiveness.

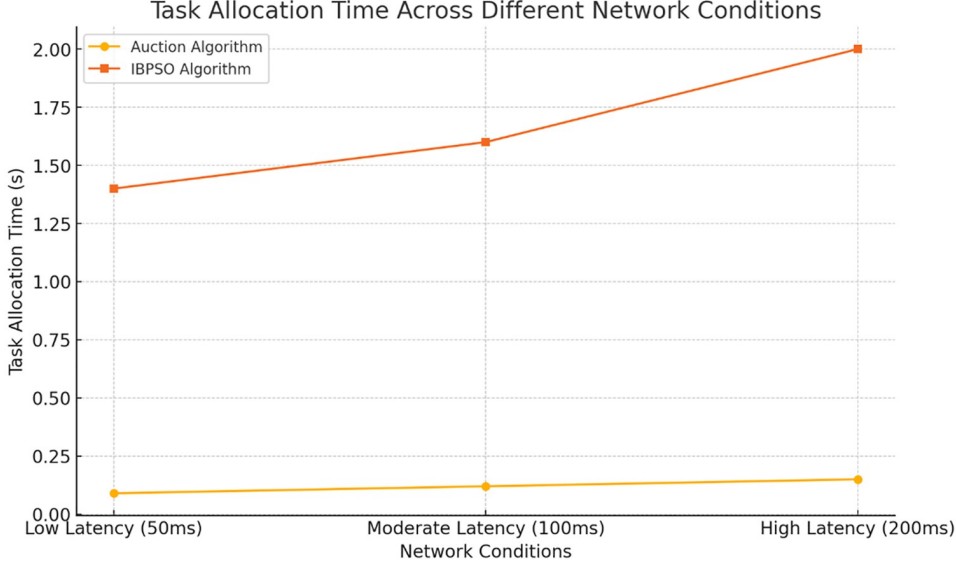

**Fig 5. Task allocation time across different network conditions.** Description: This figure illustrates how task allocation time changes under different network conditions (low, moderate, and high latency). The auction algorithm is shown to perform faster than IBPSO under all network conditions, particularly in low-latency environments.

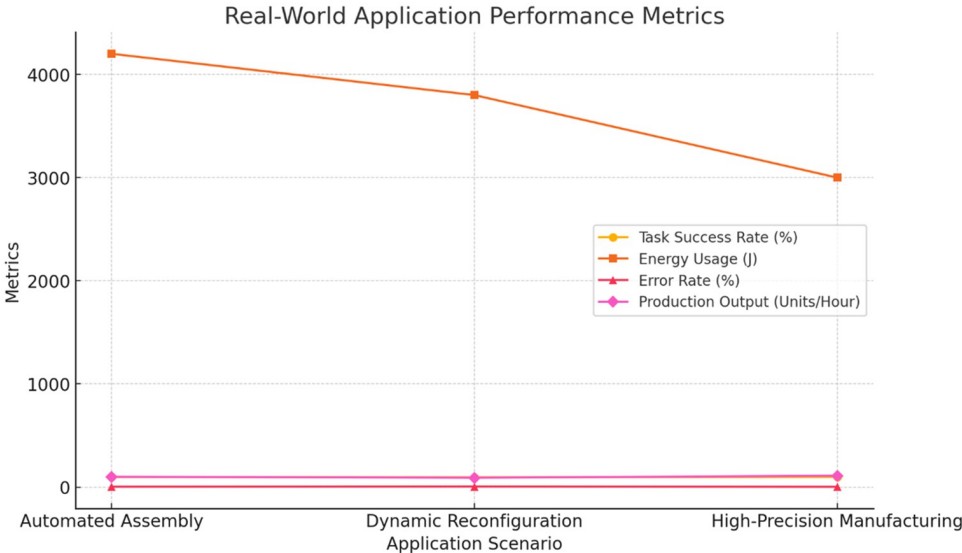

**Fig 6. Real-World application performance metrics.** Description: This figure shows the performance of the proposed algorithms in real-world scenarios, highlighting high task success rates, improved energy efficiency, reduced operational costs, and increased production output across various manufacturing applications.

## Section 4.3: Limitations and challenges

While the proposed methods do provide great potential for improving the personal AI assistant, certain setbacks should be evaluated and incorporated into further research. An important drawback is the model of the environment, which assumes the static one to be, that is not the case in nature. In factories that are smart in real life, all sorts of dynamics are present, such as layout changes, task requirements, and robot availability. The existing algorithms might need to be further developed to accommodate the dynamic nature of the environment more suitably, perhaps including the possibility of combining real-time learning and adaptations as additional functionalities.

Despite scalability, which the proposed methods will utilize, when used in the extremely large-scale manufacturing landscapes, it will have its own downsides. Despite the algorithms'

**Table 6. Real-world application scenarios and performance metrics.**

| Application Scenario | Algorithm | Task Success Rate (%) | Energy Usage (J) | Error Rate (%) | Operational Cost (USD) | Production Output (Units/ Hour) |
|---|---|---|---|---|---|---|
| Automated Assembly Line | Proposed Auction | 97 | 4200 | 3 | 2000 | 100 |
| | IBPSO | 95 | 4600 | 4 | 2200 | 95 |
| Dynamic Production Line Reconfiguration | Proposed Auction | 95 | 3800 | 5 | 2500 | 90 |
| | IBPSO | 92 | 4300 | 6 | 2700 | 85 |
| High-Precision Manufacturing | Proposed Auction | 98 | 3000 | 2 | 1800 | 110 |
| | IBPSO | 96 | 3400 | 3 | 2000 | 105 |

Description: This table provides performance metrics for different real-world application scenarios (automated assembly line, dynamic production line reconfiguration, and high-precision manufacturing). The proposed auction and IBPSO algorithms show strong performance, with high task success rates and energy efficiency across all scenarios.

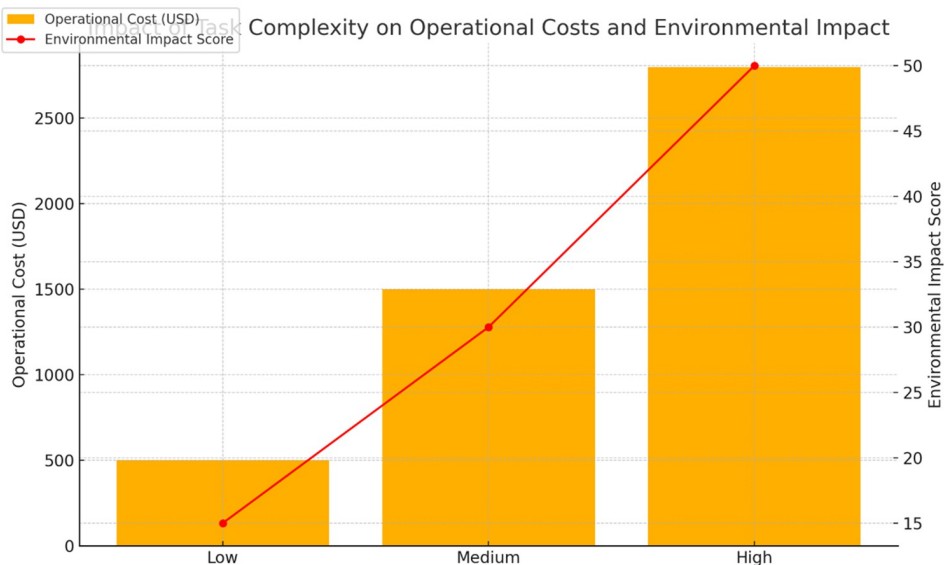

**Fig 7. Impact of task complexity on operational costs and environmental impact.** Description: The figure depicts how increasing task complexity affects operational costs and environmental impact. Higher complexity tasks increase both costs and environmental impact, but the proposed algorithms reduce these effects compared to traditional methods.

great success in simulation tests of specially chosen areas, containing a limited number of tasks and robots, the impact on environments with thousands of tasks and hundreds of robots is yet to be completely examined (Fig 7). Future research should mainly be guided by the search for faster and more effective deployments of these techniques in practical situations, as manufacturing systems get bigger and bigger.

Integrated into this is the fact that communication networks need to be reliable and consistent. In fact, symmetrical communication between the robots is needed to successfully implement auction and IBPSO algorithms due to their decentralized structure. Nonetheless, real-world implementations may face network outages or failures, and that would affect the performance of those methods. Trying out hybrid solutions, which use both centralized and decentralized task allocation, may be a way out, as they will enable the processes to fall back into action in case the communication system goes down.

Artificial intelligence and machine learning, among other technologies, could be included in such task allocation mechanisms, which would help solve the above-mentioned problems, especially in terms of flexibility to changing conditions and scalable to big systems. Therefore, the system will not only have experiences of the previous allocations, but also keep updating its knowledge, increasing its efficiency and adjustability in complex manufacturing environments.

## Section 4.4: Managerial implications

This research provides several managerial implications for industries implementing multi-robot systems in smart factories under Industry 4.0. First, the proposed decentralized auction-based task allocation algorithm allows managers to efficiently allocate resources in real-time, reducing downtime and optimizing productivity. By utilizing the IBPSO algorithm, managers can balance operational efficiency with energy consumption, aligning production

processes with sustainability goals. Furthermore, the scalability of the system ensures that organizations can adapt the solution as production demands fluctuate, enhancing long-term operational flexibility. Lastly, the insights from this study can guide decision-makers in prioritizing investments in automation technologies, leading to more informed and strategic decision-making regarding technology adoption and workforce planning in a highly automated environment.

## Section 4.5: Comparison with existing distributed algorithms

In the field of multi-agent systems, distributed algorithms such as the consensus protocol proposed in 'A novel consensus protocol using facility location algorithms' (2009 IEEE Control Applications, CCA & Intelligent Control, ISIC) have been widely applied. This approach focuses on distributing autonomous agents, such as vehicles, to achieve optimal coverage while minimizing costs using facility location algorithms. The primary advantage of such algorithms is their ability to operate efficiently in dynamic environments with localized optimization and minimal communication overhead.However, our proposed decentralized auction-based approach offers several key advantages. First, while consensus-based protocols optimize global objectives, our approach prioritizes real-time task allocation and scalability, ensuring that tasks can be dynamically reassigned as the environment changes. Second, our use of the Improved Binary Particle Swarm Optimization (IBPSO) algorithm enhances energy efficiency, which is a critical factor often overlooked in traditional consensus-based methods. On the other hand, consensus-based approaches may perform better in scenarios where minimizing communication cost is paramount. However, our approach outperforms in scalability and adaptability, especially in rapidly evolving industrial settings, where the need for real-time decision-making and energy optimization is crucial.

## Section 4.6: Comparison with distributed energy resource management techniques

Energy-efficient resource allocation is a critical aspect of distributed systems, as highlighted in recent studies such as 'Distributed energy resource management: All-time resource-demand feasibility, delay-tolerance, nonlinearity, and beyond' (IEEE Control Systems Letters, 2023). This work addresses the challenge of managing energy resources using distributed consensus-based algorithms that accommodate nonlinear dynamics and time-varying delays. By localizing decision-making, these techniques eliminate the need for centralized coordination, which can enhance system robustness and reduce communication overhead.

In comparison, our approach also emphasizes energy efficiency but applies it specifically within the context of multi-robot task allocation in smart factories. Unlike the consensus-based resource management systems that focus on the feasibility of resource demands and tolerate delays, our method leverages the Improved Binary Particle Swarm Optimization (IBPSO) algorithm to dynamically allocate tasks based on real-time energy consumption data, aiming for optimal energy use without compromising task completion rates. One advantage of our approach is its adaptability to rapidly changing industrial environments, where task demands, and energy availability may fluctuate unpredictably. However, our method may require more sophisticated energy monitoring systems compared to the consensus-based approaches, which can operate with less frequent data updates. Thus, while consensus-based methods offer robustness in handling delays and nonlinearity, our approach provides a more dynamic and responsive solution to energy management in highly automated and variable manufacturing settings.

### Section 4.7: Computational complexity

In large-scale multi-robot systems, computational complexity is a critical factor that can significantly impact the feasibility of deploying consensus-based algorithms. The proposed decentralized auction-based task allocation method, which incorporates the IBPSO algorithm, operates with a computational complexity of polynomial order. Specifically, the complexity is primarily governed by the number of robots and the number of tasks to be allocated, as well as the iterations required for the IBPSO to converge.

In comparison to traditional centralized methods, our decentralized approach allows for localized computation, which reduces the overall computational burden and makes the system scalable. This characteristic is particularly important in large-scale setups, where the complexity of the system grows with the addition of more robots and tasks. While consensus-based algorithms typically have complexities that depend on the communication network and synchronization overhead, our approach maintains efficiency through parallel task execution and optimized task allocation, ensuring that computational load remains manageable even in large-scale environments.

### Section 4.8: Communication network and quantization assumptions

In distributed robotic systems, the communication network plays a critical role in the performance and scalability of task allocation algorithms. In our research, we assume a fully connected communication network among the robots, where each robot can directly communicate with others without significant delays or packet loss. This assumption simplifies the design and analysis of the task allocation algorithm, ensuring that all robots are synchronized during task assignment and execution. Regarding quantization, our study does not explicitly model the quantization of data in information exchange or processing. However, we acknowledge that in real-world applications, the precision of data exchange can be limited due to quantization effects in both the communication network and the data processing modules. Future work can focus on incorporating quantization effects to better model realistic scenarios, where limited bandwidth and noise may affect the accuracy and timing of information exchange. This would provide a more comprehensive evaluation of the algorithm's performance in practical deployments.

### Section 4.9: Main algorithm: Decentralized task allocation with IBPSO

### Algorithm 1: Decentralized task allocation using IBPSO

\nAlgorithm 1: Decentralized Task Allocation using IBPSO\nInput: Set of robots R = {r1, r2,. . ., rn}, Set of tasks T = {t1, t2,. . ., tm}\nOutput: Optimized task allocation for robots\n\n1. Initialize particle positions and velocities for each robot\n2. Evaluate the fitness of each robot based on task completion and energy consumption\n3. While (termination criteria not met):\n a. Update particle velocities and positions using IBPSO\n b. Evaluate the fitness of new positions\n c. Update personal and global best positions\n4. Assign tasks to robots based on the optimized positions\n5. Return the final task allocation\n.

### Section 5: Conclusion

### Section 5.1: Summary of contributions

As a result, this research paper documents the enhancement of multi-robot systems, especially considering the growing importance of smart factories. Two fundamental allocation

mechanisms were developed: the first one is a distributed auction algorithm for straightforward jobs, and the second one is IBPSO, an Improved Binary PSO algorithm for complex task coalition formation.

The experiment using a distributed auction demonstrated an impressive level of scalability and resilience, which meant that it could successfully allocate robotic tasks in the highly heterogeneous and dynamic environment of manufacturing lines. Moreover, the factor that the system is decentralized makes it resilient to failures and is modifiable to accommodate diverse task requests [36, 37].

Instead of using a typical centralized PSO approach, the complexity of multiple robotics collaboration tasks was dealt with by a decentralized IBPSO algorithm [38, 39]. Utilizing coalition optimization and being more energy-efficient were not only proved to speed up the work but also became the key factor of modern smart factory operation.

We validated in experiments that the constructs achieve superior performance in terms of completion rate and computational workload in comparison to centralized methods and heuristics. Hence, the contributions of this work are twofold: designing scalable, decentralized task allocation algorithms capable of optimizing industrial processes; and provisioning energy-efficient strategies for cooperative formations, which follows the growing concern for sustainability within industry automation.

Thus, these contributions have created a strong basis for the coming intelligent production systems, allowing for solutions that are feasible and applicable in actual specific smart factories.

### Section 5.2: Future directions

Undoubtedly, the published approaches highlight the obvious progress; at the same time, there are many options that can still be discussed to make the methods more applicable and effective. A central strategy can be the expansion of these algorithms to apply to operations in environments that exhibit high variabilities of behavior. In most of the real-time smart factories, layouts, job details, and availability of robots often do change frequently.

More research work may address machine learning techniques that enable the system to learn from the previous task allocations and to adapt according to the current condition in real time by following the earlier path. This may be realized first through the establishment of predictive systems based on statistical modeling to foresee changes and proactively adjust [40].

On top of that, it is also worth mentioning that in the future, research on the scalability of algorithms in extremely large-scale applications needs to be carried out. With the manufacturing systems becoming more complex over time, a convenient metric, which can evaluate the capacity of the proposed solutions for maximizing the number of tasks and robots, remains critical [41]. It could also contain the optimization of the algorithm computational processes or develop different hybrid strategies that mix the strength between the centralized and decentralized approaches [42].

Furthermore, the vulnerability of consistent communication networks to these systems remains the potential problem that the real-world applications of these methods encounter. Researching strong communication protocols as well as designing back-up plans which can ensure system functionality during disconnections could go a long way in ensuring the resilience of the algorithms to network failures. Finally, future studies might investigate how the suggested methods for task allocation can work in tandem with cloud robotics and edge computing, which could offer additional processing power and further increase the system's resilience and scalability.

### Section 5.3: Impact statement

This paper's innovations have the potential to drastically impact the way smart industry and manufacturing operate in the foreseeable future. This study is practical in its suggestions because it tackles the most significant challenges inherent to the mixed task allocation that involve heterogeneous multi-robot systems, which can enhance efficiency, scalability, and sustainability of manufacturing systems. The introduction of energy-efficient coalition formation strategies is an occasion for industrial operations that correlates with global efforts to accomplish energy saving and reduce the environmental footprint.

The methods not only bring innovative solutions to the academic field of multi-robot task allocation but provide concrete adds for business in practice. As smart factories embrace progress, that demand for dynamic, resilient, and energy-saving task distribution mechanisms becomes ever sharper and unyielding. The algorithms used in this study are ideally suited to address these demands as they provide the basis upon which future higher-level inventions in smart manufacturing may be established.

To sum up, this study opens the avenue for the advent of more intelligent, frugal, and eco-friendly manufacturing processes. The contributions made here are expected to have a lasting impact on the field, guiding future research and development efforts and ultimately contributing to the broader goal of creating more advanced and sustainable industrial ecosystems.

## Author Contributions

**Investigation:** Zhaobin Li.

**Methodology:** Zhaobin Li.

**Supervision:** Tang Wai Fan.

**Validation:** Tang Wai Fan, Lam Sui Kei.

**Visualization:** Lam Sui Kei.

**Writing – original draft:** Qingwen Li.

**Writing – review & editing:** Qingwen Li.

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
