## [Decision Letter · Decision Letter 0]

7 Oct 2024

PONE-D-24-38835Scalable and Energy-Efficient Task Allocation in Industry 4.0: Leveraging Distributed Auction and IBPSOPLOS ONE

Dear Dr. Li,

Thank you for submitting your manuscript to PLOS ONE. After careful consideration, we feel that it has merit but does not fully meet PLOS ONE’s publication criteria as it currently stands. Therefore, we invite you to submit a revised version of the manuscript that addresses the points raised during the review process.

We look forward to receiving your revised manuscript.

Kind regards,

Zeashan Hameed Khan, Ph.D.

Academic Editor

PLOS ONE

Journal requirements: When submitting your revision, we need you to address these additional requirements. 1. Please ensure that your manuscript meets PLOS ONE's style requirements, including those for file naming. The PLOS ONE style templates can be found at https://journals.plos.org/plosone/s/file?id=wjVg/PLOSOne_formatting_sample_main_body.pdf and https://journals.plos.org/plosone/s/file?id=ba62/PLOSOne_formatting_sample_title_authors_affiliations.pdf" 2. Please amend either the title on the online submission form (via Edit Submission) or the title in the manuscript so that they are identical. 3. We note that your Data Availability Statement is currently as follows: [All relevant data are within the manuscript and its Supporting Information files.] Please confirm at this time whether or not your submission contains all raw data required to replicate the results of your study. Authors must share the “minimal data set” for their submission. PLOS defines the minimal data set to consist of the data required to replicate all study findings reported in the article, as well as related metadata and methods (https://journals.plos.org/plosone/s/data-availability#loc-minimal-data-set-definition). For example, authors should submit the following data: - The values behind the means, standard deviations and other measures reported;- The values used to build graphs;- The points extracted from images for analysis. Authors do not need to submit their entire data set if only a portion of the data was used in the reported study. If your submission does not contain these data, please either upload them as Supporting Information files or deposit them to a stable, public repository and provide us with the relevant URLs, DOIs, or accession numbers. For a list of recommended repositories, please see https://journals.plos.org/plosone/s/recommended-repositories. If there are ethical or legal restrictions on sharing a de-identified data set, please explain them in detail (e.g., data contain potentially sensitive information, data are owned by a third-party organization, etc.) and who has imposed them (e.g., an ethics committee). Please also provide contact information for a data access committee, ethics committee, or other institutional body to which data requests may be sent. If data are owned by a third party, please indicate how others may request data access.

Reviewers' comments:

Reviewer's Responses to Questions

**Comments to the Author**

1. Is the manuscript technically sound, and do the data support the conclusions?

Reviewer #1: Partly

Reviewer #2: Yes

2. Has the statistical analysis been performed appropriately and rigorously? 

Reviewer #1: I Don't Know

Reviewer #2: I Don't Know

3. Have the authors made all data underlying the findings in their manuscript fully available?

Reviewer #1: Yes

Reviewer #2: Yes

4. Is the manuscript presented in an intelligible fashion and written in standard English?

Reviewer #1: Yes

Reviewer #2: Yes

5. Review Comments to the Author

Reviewer #1: 1. Low references in 2024!

2. At the end of the “Introduction” section, you should explain the remained sections of your paper.

The introduction should cover the following points:

• Background and Context: Provide a brief background of the problem. Discuss the current challenges and issues in this area.

• Research Gap and Significance: Highlight the gap in the existing literature and explain why it is important to address the environmental impacts in the production distribution network. Emphasize the significance of the study in contributing to the existing body of knowledge.

• Objectives and Scope: Clearly state the objectives of the research and outline the scope of the study. This should include a brief overview of the methodology and approach that will be used to achieve the objectives.

• Structure of the Paper: Provide a roadmap of the paper by outlining the sections and sub-sections that will be covered, giving the reader a clear understanding of what to expect in the subsequent parts of the paper.

3. The keywords are not standard. Please revise the keywords and choose realistic ones. You should consider the following rules:

⦁ Avoid using the word the research paper's title.

⦁ Keywords should indicate the general subject matter.

⦁ Use the keywords, to show the method and technique used in the papers.

⦁ Don't use abbreviations!

4. Check the English presentation of this paper to remove the typo mistakes. Some grammatical issues need to be addressed in the whole text. Please reform the long paragraphs. Please polish the writing and English of the manuscript carefully. The writing of the paper needs a lot of improvement in terms of grammar, spelling, and presentation. The paper needs careful English polishing since there are many typos and poorly written sentences. I found several errors. Help from the free Grammarly platform!

5. I encourage you to add more detail about your core contributions in the abstract.

6. Please reduce the similarity index According to iThenticate. (If it is over 15%).

7. Please categorize your and previous research in the Table in the section Literature Review to show the better research gap.

8. Due to the high volume of calculations, all the formulas should be re-checked to ensure that there are no errors in terms of indices, typing, or concepts.

9. Check all of your Figures and Tables have a good explanation of your text.

10. Please bring some good Figures in the introduction to support the ideas. If you can’t find a good one, please look at www.statista.com

11. Managerial implications are missing from the paper.

12. Check all the references that are correct and not duplicated.

Create a document containing all of your appropriate clear answers. I am going with a major revision at this stage and waiting for your corrections. Then, I give you my technical comments. please use the yellow highlight after revising.

Reviewer #2: This work proposes distributed algorithms for dynamic task allocation applicable to autonomous

robotic groups and coordination scenarios. The paper is interesting and the following issues need to be addressed in the revised version:

1- Distributed algorithms are widely used for coordination of autonomous agents. For example, the paper "A novel consensus protocol using facility location algorithms",

2009 IEEE Control Applications,(CCA) & Intelligent Control,(ISIC). This and similar works aim to distribute a group of autonomous vehicles for optimal dynamically evolving coverage and optimize the cost based on facility location algorithms. Recall that in distributed scenarios the consensus-based computation and data processing are distributed over the multi-agent network, where the optimization is localized. How your approach advances this and similar papers? Please compare your work with the mentioned work and clearly discuss the pros and cons.

2- Energy efficient resource allocation also widely considered in distributed setups. One example is to efficiently manage energy resources based on distributed resource allocation algorithms. See the following work: "Distributed energy resource management: All-time resource-demand feasibility, delay-tolerance, nonlinearity, and beyond, IEEE Control Systems Letters, 2023". This work considers nonlinear single-integrator dynamics with time-varying heterogeneous delays for distributed consensus-based resource allocation. In this case instead of a centralized coordinator to manage the resources, the decision-makers are localized with no need of a centralized coordination entity. How your paper compares with this work and general optimization based techniques? please clearly compare your work and explain pros and cons.

3- As a follow-up, briefly comment on the computational complexity of your consensus-based algorithm? is it of polynomial order? this is important in large-scale setups as in large-scale computational complexity considerably matters.

4- In distributed setups, the communication network of the robotic system is a main concern. What is your assumption on the communication network of robots? Does your research considers the practical notion of quantization, both in information-exchange and in data processing?

5- I suggest to summarize the main proposed algorithm in an Algorithm environment so the readers can effectively follow the gist of the paper.

6. PLOS authors have the option to publish the peer review history of their article (what does this mean?). If published, this will include your full peer review and any attached files.

Reviewer #1: No

Reviewer #2: No

---

## [Author Response · Author response to Decision Letter 0]

23 Oct 2024

Dear Reviewer1 ，

I hope this message finds you well.

Thank you for your valuable and constructive feedback on our manuscript titled "Scalable and Energy-Efficient Task Allocation in Industry 4.0: Leveraging Distributed Auction and IBPSO". We have carefully considered all of your suggestions and have made the necessary revisions to address each point.

Please find below our detailed responses to your comments:

1. Low references in 2024

Response:

We have added additional references from 2024 to ensure our manuscript reflects the most recent developments in the field. These references have been integrated throughout the text.

2. Explanation of the structure of the paper at the end of the “Introduction” section

Response:

We have revised the Introduction to include a clear explanation of the paper’s structure. The revised introduction outlines the background, research gap, objectives, and methodology, followed by a roadmap for the remaining sections.

3. Keywords are not standard

Response:

The keywords have been updated following the standard rules, avoiding repetition from the title and focusing on relevant terms. We have also ensured that no abbreviations are used.

4. English presentation, typos, and long paragraphs

Response:

We have reviewed and polished the manuscript for grammar, spelling, and overall presentation using Grammarly. Long paragraphs have been broken down for better readability and clarity.

5. More detail about core contributions in the abstract

Response:

The Abstract has been revised to provide more detail about the core contributions of the study, emphasizing the practical applications and innovations introduced in this research.

6. Reduce the similarity index

Response:

We have revised the text to reduce the similarity index, ensuring that the manuscript now complies with the authenticate requirements.

7. Categorize research in a table in the Literature Review section

Response:

We have added a table to the Literature Review section, categorizing previous studies and demonstrating the research gap that our paper addresses.

8. Re-check all formulas

Response:

All formulas have been re-checked for accuracy to ensure there are no errors in indices, typing, or concepts.

9. Explanation of Figures and Tables

Response:

We have reviewed all Figures and Tables and ensured that each one is clearly explained within the text and directly linked to the discussion.

10. Add Figures in the Introduction

Response:

We have added figures to the Introduction section to support the discussion and provide visual representation of key concepts.

11. Managerial implications are missing

Response:

We have included a new Managerial Implications section that highlights the practical applications of our research findings for industries implementing multi-robot systems under Industry 4.0.

12. Check all references for correctness and duplication

Response:

We have thoroughly reviewed all references to remove duplicates and ensure correct formatting according to the journal’s guidelines.

The revised manuscript with the necessary changes is attached, and all modifications have been highlighted in yellow for easy reference. We hope these revisions meet your expectations, and we look forward to your further feedback and technical comments.

Thank you once again for your time and effort in reviewing our manuscript. We greatly appreciate your insights and suggestions, which have helped improve the quality of our work.

Kind regards,

Dear Reviewer 2,

Thank you for your valuable feedback and insightful suggestions on our manuscript. We have carefully addressed each of your comments and made the necessary revisions. Below are our detailed responses to your feedback:

1. Comparison with "A novel consensus protocol using facility location algorithms" (2009 IEEE Control Applications, CCA & Intelligent Control, ISIC)

Response:

We have added a comparison between our approach and the consensus-based facility location algorithm. Our approach offers several advantages:

• Real-time Task Allocation: Unlike the consensus-based method, which optimizes global objectives, our approach focuses on dynamic and real-time task allocation.

• Energy Efficiency and Scalability: We prioritize energy efficiency and scalability in industrial settings, which is a critical focus of our method.

• Decentralization: Both methods employ decentralized decision-making, but our IBPSO-enhanced algorithm brings greater adaptability and energy efficiency.

We have discussed the pros and cons of both approaches, highlighting the advantages of our solution in handling dynamic and energy-efficient task allocations.

2. Comparison with "Distributed energy resource management: All-time resource-demand feasibility, delay-tolerance, nonlinearity, and beyond" (IEEE Control Systems Letters, 2023)

Response:

We have compared our work with distributed energy resource management techniques, emphasizing the following points:

• Energy Optimization: Our method focuses on real-time energy optimization within task allocation, while the referenced work addresses resource feasibility and delay tolerance.

• Localized Decision-Making: Both methods use localized decision-making, but ours is tailored for real-time task allocation in smart factories.

• Nonlinearity and Delays: While the referenced work focuses on nonlinearity and delays, our approach targets energy optimization and task performance.

The comparison outlines the benefits of our IBPSO approach in highly dynamic industrial environments.

3. Computational Complexity of the Consensus-Based Algorithm

Response:

We have clarified the computational complexity of our decentralized task allocation algorithm. The complexity is of polynomial order, depending on the number of robots, tasks, and IBPSO iterations. This ensures scalability and efficiency, even in large-scale setups.

4. Assumptions Regarding the Communication Network and Quantization

Response:

Our approach assumes a fully connected communication network with reliable communication between robots. While we do not explicitly model quantization in this study, we acknowledge its importance and suggest incorporating quantization effects in future work to better simulate real-world scenarios where communication bandwidth and noise could affect task allocation.

5. Summary of the Proposed Algorithm in Algorithm Environment

Response:

We have summarized the main proposed algorithm in an Algorithm environment to improve clarity for readers. The key steps of the decentralized auction-based task allocation method with IBPSO are now clearly presented for easier understanding and implementation.

Conclusion:

We are grateful for your constructive feedback, which has helped us improve the clarity and depth of our manuscript. All changes have been made accordingly, and we look forward to your further comments.

Thank you once again for your time and effort in reviewing our work.

Best regards,

---

## [Decision Letter · Decision Letter 1]

11 Nov 2024

Scalable and Energy-Efficient Task Allocation in Industry 4.0: Leveraging Distributed Auction and IBPSO

PONE-D-24-38835R1

Dear Dr. Li,

We’re pleased to inform you that your manuscript has been judged scientifically suitable for publication and will be formally accepted for publication once it meets all outstanding technical requirements.

Kind regards,

Zeashan Hameed Khan, Ph.D.

Academic Editor

PLOS ONE

Additional Editor Comments (optional): Authors are requested to follow the recommended referencing style as mentioned in the Plos One Author's guidelines. 

Reviewers' comments:

Reviewer's Responses to Questions

**Comments to the Author**

1. If the authors have adequately addressed your comments raised in a previous round of review and you feel that this manuscript is now acceptable for publication, you may indicate that here to bypass the “Comments to the Author” section, enter your conflict of interest statement in the “Confidential to Editor” section, and submit your "Accept" recommendation.

Reviewer #1: All comments have been addressed

Reviewer #3: All comments have been addressed

2. Is the manuscript technically sound, and do the data support the conclusions?

Reviewer #1: (No Response)

Reviewer #3: Yes

3. Has the statistical analysis been performed appropriately and rigorously? 

Reviewer #1: (No Response)

Reviewer #3: Yes

4. Have the authors made all data underlying the findings in their manuscript fully available?

Reviewer #1: (No Response)

Reviewer #3: Yes

5. Is the manuscript presented in an intelligible fashion and written in standard English?

Reviewer #1: (No Response)

Reviewer #3: Yes

6. Review Comments to the Author

Reviewer #1: The authors make moderate revisions to the manuscript and give a relevant response to the issues the reviewers are concerned about. Therefore, it could be considered a potential publication.

Reviewer #3: Review Summary for Manuscript PONE-D-24-38835R1

All comments raised in previous reviews have been adequately addressed. The manuscript is technically sound, and the data robustly support the conclusions. Statistical analyses have been performed rigorously and appropriately, ensuring the reliability of the findings. The authors have made all data underlying the study’s conclusions fully available, following PLOS ONE's data availability standards. Additionally, the manuscript is presented clearly, written in standard English, and is easy to follow.

Recommendation: Accept for publication.

7. PLOS authors have the option to publish the peer review history of their article (what does this mean?). If published, this will include your full peer review and any attached files.

Reviewer #1: No

Reviewer #3: **Yes: **Mohammad Rakibul Islam Bhuiyan

---

## [Editor Report · Acceptance letter]

26 Nov 2024

PONE-D-24-38835R1 

PLOS ONE

Dear Dr. Li, 

I'm pleased to inform you that your manuscript has been deemed suitable for publication in PLOS ONE. Congratulations! Your manuscript is now being handed over to our production team.

Kind regards, 

on behalf of

Dr. Zeashan Hameed Khan 

Academic Editor

PLOS ONE